# Confronting *Mean Girls* Niceness: Conceptualizing Whisper Care to Disrupt the Politics of Niceness in Academia

Rose Ann E. Gutierrez [1,*], Carolyn S. F. Silva [2] and Ruby Batz [2]

1 Department of Educational Studies, University of Nevada, Reno, NV 89557, USA
2 Department of Education Preparation, University of Nevada, Reno, NV 89557, USA; carolynsilva@unr.edu (C.S.F.S.); rbatzherrera@unr.edu (R.B.)
* Correspondence: roseanng@unr.edu

**Abstract:** While the current literature on Niceness in higher education has examined the discourses and practices of Niceness in academic spaces, making it more identifiable, less is known about how minoritized faculty navigate and disrupt the culture of Niceness. The purpose of this article is to offer a resistance-based framework to combat academia's Niceness culture through the lens of the authors. Using theory in the flesh as theory and methodology, we use collaborative autoethnography to conceptualize Whisper Care to give language and articulate an orientation and philosophy rooted in Kindness. Our findings present a process to confront Niceness while guiding, supporting, and protecting each other in higher education institutions. We conclude with implications for future research and practice for faculty and higher education leaders.

**Keywords:** Niceness; Kindness; whiteness; higher education; collaborative autoethnography

## 1. Introduction

Regina: Oh my god! I love your skirt. Where did you get it?

Lea: Uh, it was my mom's in the 80s.

Regina: Huh! Vintage! So adorable.

Lea: Thanks! [smiles and walks away]

Regina to Cady: That is the ugliest effing skirt I've ever seen.

Tina Fey's mom inspired this iconic scene from the movie *Mean Girls*. In a 2004 interview, Fey elaborated on the context that inspired this scene from the movie. Fey revealed, "My mom has this habit that if she sees a lady in a really ugly hat or a glittery sweatshirt, she'll go, 'I love your shirt,' and I'll say, 'Mom, that's really mean,'" [1]. This quote about Fey's mom offers a glimpse into the coded language of Niceness, which on the surface sounds polite, pleasant, and nice, despite its passive-aggressive undertones. Fey explicitly states that this type of interaction is a "habit" for her mom, a regular practice of complimenting another individual, disguised by Niceness, coded to express an insult. Beyond illustrating the relational aspects of Niceness and its impacts on daily interactions, the dialogue with Regina from *Mean Girls* hints at the embodied manifestations of Niceness historically shared and enacted by White women at higher education institutions [2]. When Niceness is examined as an analytical category, the rhetoric and practices of Niceness reveal a more insidious emic with familial connections to whiteness that reproduces injury and harm onto others, which is especially prevalent in education spaces [3].

While the current literature on Niceness in higher education has examined the discourses and practices of Niceness in academic spaces, making it more identifiable, less is known about how minoritized faculty navigate and disrupt the culture of Niceness. The purpose of this article is to offer a resistance-based framework to combat academia's

Niceness culture through the lens of the authors. In doing so, we draw from feminist onto­epistemologies of carework and give language to articulate an orientation and philosophy rooted in Kindness. We define Kindness as an approach to life in academia that prioritizes humanized interpersonal relationships rooted in ethical, respectful, and meaningful con­nections between like-minded people who seek to build a more just academic world. We conceptualize the term Whisper Care as a form of armor to confront U.S. American Niceness while guiding, supporting, and protecting each other in higher education institutions. The following question guides this article: How do three-tenure track faculty members who identify as minoritized women embody a praxis of carework while navigating academia through Kindness?

We divide this article into four parts. First, we provide an overview of the literature to draw explicit connections between Niceness in academia and the experiences of minoritized faculty in violent academic environments. Second, we describe our conceptual framework and use of theory in the flesh as both theory and methodology within our inquiry. In this section, we detail our use of collaborative autoethnography (CAE) as a method of collecting data, processes of coding and analysis, and positionalities that shaped the research process and the writing of this article. Third, we present findings from the CAE and, more specifically, elaborate on our conceptualization of Whisper Care rooted in our lived experiences as data. Lastly, we discuss our findings in relation to the current literature on Niceness in academia as experienced by minoritized faculty and conclude with implications for future research and practice in higher education.

## 2. Literature Review

For historically minoritized faculty situated on the margins or outside of privileged social and historical locations, academia has been a hostile space designed to preserve, center, and perpetuate whiteness. These same academic spaces that harm and exclude minoritized faculty, staff, and students embrace a culture of Niceness that privileges politeness, neutrality, comfort, and civility [4]. Nevertheless, minoritized faculty have found ways to resist the pressures of Niceness culture [5,6]. To explicitly draw connections between how minoritized faculty have experienced and navigated varying challenges in academia with a focus on the discourse of Niceness in a higher education context, we divide this literature review into three parts: (1) academia and Niceness, (2) minoritized faculty experiences, and (3) whisper networks and carework.

### 2.1. Academia and Niceness

Although the pervasiveness of Niceness in academic spaces has been documented by scholars [2,4,5,7,8], as a cultural construct, U.S. American Niceness dates back to settler colonial efforts to justify the violence of colonization by positioning Americanness as in­herently nice and innocent and, therefore, incompatible with violence and negativity [9]. Ideas of Niceness also served to obscure White women's active involvement in the colonial enterprise, such as their participation in the slave market—making up 40% of all slave owners—and opposition to emancipation [10,11]. Thus operationalized, Niceness has been historically employed to promote a more favorable view of whiteness as a neutral, harmless, and desirable value. With the economic changes of the 19th century and the subsequent fem­inization of the teaching profession, White women gained a foothold in the schoolhouses across the country and became the "educative arm of the empire" [12] (p. 23). This shift replaced the masculine thrust of education to introduce a more subtle form of violence that hid behind a nurturing veneer to reinforce white supremacy. In time, Niceness became a "shared socio-emotional disposition, particularly amongst White women" [13] (p. 91) who have historically monopolized the field of education by intentionally excluding Women of Color from education opportunities through a combination of White liberal feminist values and weaponized Niceness. Unsurprisingly, Niceness is embedded in all domains of U.S. education, from curriculum to pedagogy and faculty hiring processes, shielding White educators from acting upon oppressive structures and practices [3,7,13,14]. This seemingly

well-intentioned culture of Niceness hides the historical and strategic project of whiteness that shapes educational experiences, limits discussions of race and racism, and reinforces white supremacy [15]. In academia, Niceness also acts as a proxy for whiteness by promoting narratives of individualism and meritocracy, disavowing and reifying structural, historical, and social patterns of oppression [3], and creating a culture that "(re)produces toxic, exclusionary, and inequitable environments" [5] (p. 7).

*2.2. Minoritized Faculty Experiences*

Though the underbelly of whiteness that feeds Niceness culture often goes unnamed in academic spaces, its material effects deeply impact the experiences of minoritized faculty who are systematically marginalized, isolated, and pressured to remain silent on issues of racial discrimination and inequity [8,16–18]. Beyond the emotional and social impacts of Niceness culture, minoritized faculty also struggle with epistemic exclusion or the marginalization of scholars and scholarship that falls outside of white-centric knowledge and practices [19]. Because Niceness culture is predicated on racialized inequity and works to maintain white comfort, minoritized faculty are disproportionately assigned mentoring and service responsibilities in comparison to White faculty; their research—often focused on the needs of marginalized groups and non-traditional frameworks—is undervalued and more harshly evaluated by administrators [20–23]. This shields White faculty from acknowledging and addressing racism, white privilege, and other unequal power dynamics, effectively preventing minoritized faculty from advocating for a more equitable work environment. Under these dynamics, minoritized faculty who fail to comply with the norms of Niceness are often seen as troublemakers and left vulnerable to retaliation [24]. Subjected to unequal systems of evaluation rooted in racialized biases under the guise of race-neutral and colorblind language, minoritized faculty often experience disparities in performance evaluations with negative consequences for productivity, academic recognition, merit, tenure, and promotion [17,25]. More specifically, minoritized women and Women of Color, are vulnerable to issues of social exclusion, belonging, and retention, which impact their career prospects and well-being [26–28]. Severely underrepresented in tenure-track positions in higher education institutions [29], these women also must contend with the invisible labor of constantly decoding and navigating the lexicon of Niceness that shapes higher education institutions in the United States.

*2.3. Whisper Networks and Carework*

The #MeToo movement in the early 2000s popularized the term "whisper network" as a phenomenon to describe informal communication (e.g., word of mouth, private online forums, crowd-sourced documents, etc.) between women who privately circulated stories of sexual harassment and violence to warn other women about potential perpetrators in their respective professional community or industry [30], including family members [31]. While most of the discourse about whisper networks is associated with the #MeToo movement, whisper networks are not exclusive to topics surrounding sexual harassment. Whisper networks within the academic context are "small groups of trusted colleagues from similarly marginalized communities with whom [they] can share struggles, strategies, and victories" [32] (p. 330). Within the past decade, scholars in various academic fields, such as anthropology [33], communication [30,34], and gender studies [32], have discussed and empirically examined the concept. While the term has recently become more recognizable in public and academic discourse, the act of sharing information between women to arm each other with information and protect one another has been a long-held practice between women within feminist knowledge production [35]. Whisper networks operate from the intellectual and political literacy practice of chisme, which historically, Women and Girls of Color have wielded as a tool to combat the intersections of racism, sexism, and classism [36]. Additionally, chisme, as an active form of storytelling, can be used as a form of healing [31], offering strategies informed by a praxis of carework. Rooted in feminist ontoepistemology, we define carework as a communal responsibility that seeks to

nurture the body/mind/spirit [37], radically commits to calling *in* (as opposed to calling *out*) others to engage in brave conversations for collective growth [38,39], and maintains a politics of refusal—a refusal to prioritize power over people [40]. In this context, carework offers minoritized women in academia an alternative to enact solidarity, affirm collective interdependence, and honor ancestral ways to connect with one another in the face of intersectional oppression and institutional carelessness perpetuated by Niceness culture [41]. For example, in academia, Faculty of Color have historically shared their experiences in trusted communities when navigating the culture of Niceness, "[including] how they relied on their collectivistic values to create supportive relationships with each other" [5] (p. 6). Within the context of whisper networks, carework manifests in the practice of sharing stories with other women for sustenance and to equip one another with information [33] to strategically navigate hostile climates in higher education institutions [32]. The function of whisper networks and carework within academic institutions is similar to that of chisme as a storytelling tool that helps women to protect each other [31]. However, the main difference between these storytelling tools is about the end goal of sharing such information. While prior conceptualizations of chisme focus on storytelling to protect relatives from potential sexual violence [31], the praxis of Whisper Care not only includes sharing a variety of information needed to navigate tenure-track journeys carefully and successfully within academic institutions, but also includes building a culture of trust and reciprocity amongst women [36].

## 3. Conceptual Framework

"A theory in the flesh means one where the physical realities of our lives—our skin color, the land or concrete we grew up on, our sexual longings—all fuse to create a politic born of necessity. Here, we attempt to bridge the contradictions in our experience... We do this bridging by naming ourselves and by telling our stories in our own words." [42] (p. 19)

### 3.1. Theory in the Flesh as Theory and Methodology

We begin our conceptual framework with an excerpt from Moraga and Anzaldúa as they call on women of all colors in academia to carve out a space in their own terms and rebel against traditional paradigms by creating theories born out of lived experience. Moraga and Anzaldúa name this process a "theory in the flesh" where they declare that theory should not solely come from academic texts, but flow from "what we live, breathe, and experience in our everyday lives... useful for liberation" [43] (p. 216). As such, for the purpose of this article, we use theory in the flesh as both theory and methodology.

As a theoretical framework, theory in the flesh provides a conceptual anchor to examine our embodied subjectivities as three tenure-track faculty women located on the margins, while providing a theoretical tool to give language to how we experience the cultural politics of emotion in academia [44]. Theory in the flesh conceptualizes the body as a material site where knowledge exists, and experiences are theorized as a consequence of shifting social and geopolitical locations [45]. Additionally, for us in this inquiry, we are informed by Black and Native feminists who conceptualize "the flesh" as a "gathering place or connective tissue" [46] (p. 9), where we can exist and recognize a shared intimacy and solidarity through our stories [47,48]. By using theory in the flesh as a theoretical framework, we can better understand how lived experience in our flesh offers a site to reckon with and discern Niceness that insidiously operates as the norm in academic spaces, where our bodies are subjected to the violent structures and harmful processes in academia [46,49]. As a methodology, theory in the flesh provides critical sites for analysis to reassociate with body/mind/spirit that have been fragmented by the colonial university [42,43,50]. Additionally, theory in the flesh offers us analytical tools to design a collaborative autoethnography with lived experience as entry points into material sites of knowledge to be analyzed and understood as data [43]. Specifically, theory in the flesh allows researchers to deconstruct systems of power in academia, reconcile the contradictions in our lived

experiences as data, and heal through the processes of meaning- and sense-making during analysis and dissemination [43,45,51].

*3.2. Unearthing Sources of Our Theory in the Flesh through Collaborative Autoethnography*

We, Rose Ann, Carolyn, and Ruby, initially met in August of 2022 as we began the 2022 academic year at the University of Nevada, Reno. Our College reconvened all faculty and staff in person, and this same year, Rose Ann and Carolyn started their first year as tenure-track assistant professors. While we three formed individual relationships with each other (i.e., Rose Ann and Ruby bonded as office neighbors; Ruby and Carolyn shared experiences being in the same department; and Rose Ann and Carolyn found affinity as first-year assistant professors), we finally all connected at the beginning of spring of 2023 over dinner. During this initial meeting, we talked for two hours about our personal lives and navigating the professoriate as first-generation faculty members located on the margins. We organically created an academic homespace, a community where we sought refuge from microaggressions in academia, a space where we can just *be*, without explanation for the other person to understand our experiences. We realized that our academic homespace serves as a counterspace [52] to both survive and thrive as assistant professors who have to crawl through an academic minefield of the culture of Niceness, and climb an uphill battle to tenure and promotion. As such, we decided to systematically document, examine, and analyze how we collectively navigated Niceness.

In alignment with our conceptual framework, we employed collaborative autoethnography to collect data for this article [53,54]. CAE is the study of two or more researchers in a particular context [55]. We use CAE to study ourselves collectively as three tenure-track faculty members to document how we experience and navigate Niceness at our institution, which "[deepended] the analytical and interpretive components" [54] (p. 598) of the inquiry. While we have engaged in informal conversations about navigating institutional challenges, building trust and a stronger bond, and calling ourselves "three peas in a pod", we began a formal process of collecting data by recording our dialogues. Our CAE process is dialogic in nature. We met via Zoom and in person several times to share challenges in the College and our departments, with a focus on navigating Niceness. To model Kindness to colleagues who, as we discussed, have verbally harmed us, we leave out events and details that can lead to the identification of others. Instead, our analysis focuses on personal memories and recollections of the process regarding how we tackled obstacles collectively. The dialogic process of engaging in meaning- and sense-making with one another challenged us to be critical, self-reflexive, and rigorous about the ways we understood and examined our lived experiences as individuals and as a collective. We recorded these sessions and transcribed them for analyses. Additionally, we collected and analyzed data in the forms of personal memory/recollection, self-observation, self-reflection, and self-analysis, including transcripts from the recorded dialogue, text messages between the researchers, journal entries, memos, and other reflective writings [53] (see Table 1).

To answer our research question that pertained to our embodied subjectivities that informed the ways we navigated Niceness in academia, we used affective methods of coding and analysis to "investigate subjective qualities of human experiences (e.g., emotions, values, conflicts, judgments) by directly acknowledging and naming those experiences" [56] (p. 105). We used affective methods of coding because our analysis was concerned with the relationship between affect, embodied knowledge, and power [57,58], and sought to examine the stories in the flesh. Affective methods of coding include emotion coding and values coding, which we used to understand the inner workings of our cognitive frames [56] as we recounted our lived experiences during the CAE in the first cycle of coding. Additionally, we used versus coding in the second cycle of coding to reckon with the tensions, conflicts, and contradictions that lived in our data between the politics of Niceness versus enacting a politics of refusal through Kindness to make sense of how our bodies and emotions were enmeshed with embodied subjectivities. Our use of an affective–discursive framework [57] during this process assisted in our analysis to deconstruct language and behavior that

lived within the stories we told ourselves and one another [59], which then helped us to decipher the coded language of Niceness versus Kindness. We each conducted individual analyses and then collectively came together as a team to systematically identify themes that emerged from our individual analyses. As a result, our analysis, shaped by a politics of necessity to bridge the fragmented scholar [42], led us to conceptualizing Whisper Care as a pedagogical act of refusal to conform and be consumed by the colonial university. Affective methods aligned with theory in the flesh as theory and methodology, since we examined our lived experience as data and, more specifically, sought to understand how bodies can be corporeal sites of knowledge, as well as how that knowledge is created, shaped, and informed by our positionalities [57].

**Table 1.** Autoethnographic data.

| Data Type | Time Focus | Source/Authorship | Example(s) |
| --- | --- | --- | --- |
| Personal memory/recollection | Past | Self | Snapshot of writings and talking story recollecting memories by researchers about experiencing and navigating Niceness rhetoric and practices |
| Self-observation | Present | Self | Self-observation logs and memos capturing researchers' streams of consciousness |
| Self-reflection | Past and present | Self | Free-form journals and self-reflective writings created by the researchers |
| Self-analysis | Past and present | Self | Self-focused writings created by the researchers with predetermined frameworks and concepts for analysis in connection with researchers' lived experiences |
| Interview | Past and present | Self-other or other | Recorded dialogue with research teammates about the topic of inquiry, which included other data types above |

Note: We adapted Chang and colleagues' table of autoethnographic data types to illustrate the data we collected and analyzed for this article [53].

### 3.3. Positionality

As critical educators and researchers hailing from the Global South—specifically, as a 1.5-generation immigrant from the Philippines and first-generation immigrants from Brazil and Guatemala, respectively—we, Rose Ann, Carolyn, and Ruby, collectively engaged in a profound exploration of our racialized identities and encounters with racism and sexism, which significantly inform and influence our approach to studying Niceness, Kindness, and Whisper Care in academia. We all share being tenure-track assistant professors at the same institution, yet our unique positions within different programs contribute diverse perspectives to our individual experiences and collaborative efforts.

Rose Ann identifies as a 1.5-generation immigrant, and as a Pinay and Woman of Color racialized as Asian in the United States. Her research examines the connections between knowledge, race, and social transformation in higher education contexts using critical theories and critical methodologies with a focus on Asian American and Pacific Islander students. Carolyn identifies as a cisgender woman who immigrated from Brazil and is racialized as Latina in the United States. Her research employs critical intersectional perspectives to understand how race, ethnicity, gender, and immigration impact the educational experiences of Latinx and AfroLatinx students and communities. Meanwhile, Ruby, an immigrant cisgender Woman of Color racialized as Latina in the United States, specializes in examining family engagement practices within early learning settings. Her focus extends to exploring the intersections of race, ethnicity, disability, and language. It is precisely the combination of our positionalities, social locations, and our theoretical and methodological orientations that gives us a shared framework in navigating and resisting white academic cultures. Our shared experiences have fostered a community of care and praxis, allowing for profound and reflexive discussions during the various stages of our research. Leveraging our diverse knowledge, we engaged in collaborative dialogues

that informed our research design, facilitated the interpretation of data, and guided the articulation of our findings.

## 4. Conceptualizing Whisper Care

We came into this inquiry based on our individual experiences entering meetings and conversations in higher education spaces and leaving those spaces "feeling some type of way". Ahmed describes this feeling as a sensation that is impactful because the feeling leaves a person with "an impression that is not clear or distinct" [60] (p. 22). She digs deeper into discussing this sensation as a "feminist gut" with "its own intelligence", and encourages an individual to "get closer to the feeling" [60] (p. 22). As we reflect on the introduction of this article (i.e., the dialogue from *Mean Girls*), for us, "feeling some type of way" in academic spaces was our bodies communicating to us that it was processing and decoding the daily practices and rhetoric of Niceness we encountered as exemplified in the dialogue between Regina and Lea from *Mean Girls*. Unless one has been trained and socialized to read the rhetoric of Niceness through daily interactions and dialogue (e.g., Tina Fey's mom), an individual can have a challenging time deciphering this coded language, which can result in emotional and psychological harm, especially for minoritized early-career scholars in academia. Once we came together and shared our various illegible encounters of Niceness through experiences that left us confused, unsettled, frustrated, irritated, or broadly "feeling some type of way", we were able to make sense and capture the politics of emotion in the flesh, thus resulting in the conceptual birth of Whisper Care.

In the spirit of theory in the flesh, we leaned "closer to the feeling" [60] (p. 22) by conducting a collaborative autoethnography to engage in the processes of bridging, reflecting, healing, and building after confronting harm in academic spaces and, more particularly, being challenged by navigating the politics of Niceness. This process allowed us to create a counterspace, an alternative academic world that could center, protect, and nourish its historically marginalized members and their ways of knowing, being, and doing. After conducting an affective–discursive analysis [57] and decoding the practices and rhetoric of Niceness versus Kindness, we identified patterns from the data that revealed a critical distinction between the two:

> Ruby: [Kindness] it's a philosophy to find and connect with people whose orientation intends to build a more just academic world. It allows us to connect with others in ethical, respectful, and meaningful ways to seed different worlds.
>
> Carolyn: Whereas Niceness is a strategy.
>
> Rose Ann: Niceness is the tool of the oppressor. But for us, we don't really use the tool. It's [being kind] the way we just are.

The dialogue above not only showcases our capacity to build together as we finish each other's sentences, but also how we differentiate Niceness "as a tool of the oppressor" and Kindness as a "philosophy" and "orientation" with the intent of creating a more "just academic world". Niceness is often superficially associated with civility, conceals deeper systemic alignments that perpetuate whiteness, and overshadows the authentic ethos of Kindness while hindering genuine inclusivity, especially for historically minoritized faculty [15]. This article, therefore, pivots towards a comprehensive exploration, delving into the lived experiences at the crossroads of academic Niceness, challenges faced by minoritized faculty, and strategic mechanisms such as whisper networks and carework. Building upon this foundation, our subsequent sections will offer deeper insights into how we conceptualize and use Whisper Care as a framework and lens, paving the way for our forthcoming findings.

Through our CAE, we came to understand and define Whisper Care as extending beyond theoretical contemplation. We fused our conceptions of the informal communication that occurs within whisper networks with the ontological orientations of carework, which we practiced with one another, to conceive Whisper Care. Collectively, we define Whisper Care as an embodied philosophy that actively revitalizes and preserves traditions,

safeguarding the shared humanity of those intentionally using it and/or being impacted by it. Embracing the ethos of Whisper Care, grounded in theorizing from the flesh, underscores a commitment to preserving the human aspects often overshadowed by traditional academic paradigms. Whisper Care aspires to shape a more just academic world as a transformative force, fostering connections that transcend individual experiences and promote ethical, respectful, and meaningful engagements. Rooted in an intention to "seed a different world", this philosophy actively contributes to creating communities of care and praxis. Whisper Care emerges as a theoretical construct and a practical approach to challenging and dismantling systemic racism within academic spheres. As a framework, Whisper Care provides a comprehensive and holistic vision for cultivating a more inclusive scholarly environment. Functioning as a lens, Whisper Care offers insight into how Niceness operates as a proxy for whiteness in academic spaces. Crucially, Whisper Care emphasizes how historically marginalized women in academia counter Niceness by prioritizing relationships and systems of care, sharing experiences and information with each other, which affirms and validates the knowledge and experiences of historically marginalized individuals. Within this process (see Figure 1), Whisper Care aims to uphold the authenticity inherent in our hearts, minds, bodies, and souls, legitimizing our lived experiences as foundational elements in academic discourse. From our personal memories of our communal collegial relationship, we stitched together a recollection of how we have been able to navigate challenges as early-career scholars together by mapping out three vital questions within the process of a Whisper Care framework below.

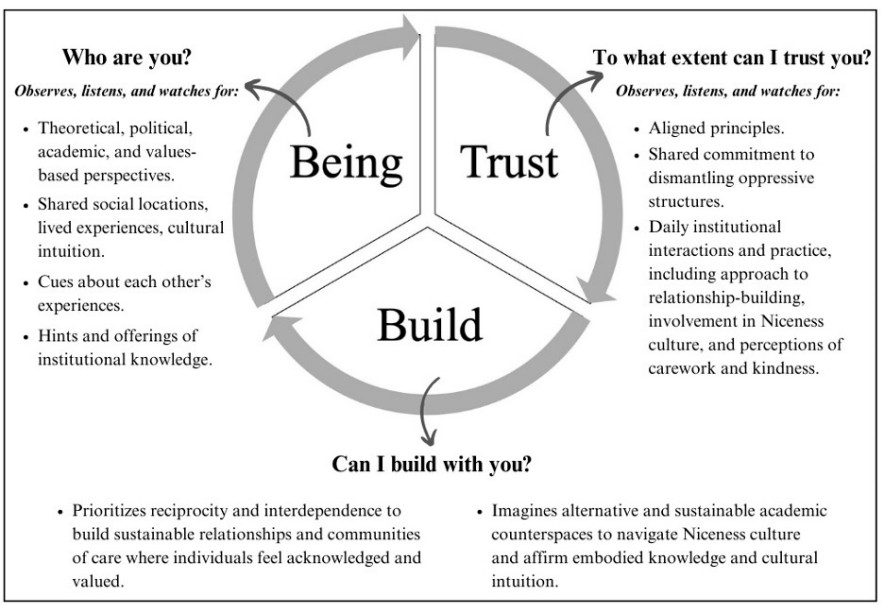

**Figure 1.** Visualizing Whisper Care as a framework.

### 4.1. Who Are You?

Our relationship began by first inquiring about one another and asking, "Who are you?" In this stage, we explored possibilities of alignment across theoretical, political, academic, and values-based perspectives. Recognizing the intricacies of social location and positionality, we draw inspiration from Zora Neale Hurston's quote, "All my skinfolk ain't kinfolk"—acknowledging that trust in one's body and instinctive Global Positioning System or GPS is essential to decode actions, ensure alignment with others within academic space, and detect possible pathways for journeying together in the future:

> Ruby: There is an exploration phase at the beginning. . . something that definitely seems like an entry point is [the person's] orientation that may be theoretical, political, epistemological. . . But to me, that orientation gives me an idea of where you are and if our values may align. . . During this stage, coding is an important

strategy because that's what we're doing all the time... using our bodies and our intuitions to decode.

Carolyn: To screen for the lexicon of whiteness, the telltale dispositions of Niceness versus Kindness, such as gaslighting versus confirming microaggressions versus microaffirmations. When these are not present, we can move forward.

From this dialogue, we learn that Whisper Care, at its foundational stage, involves intentionally listening to institutional colleagues' cues and offerings about their experiences with the institution and people from the institution. At this initial stage, academics must engage in deciphering both verbal and nonverbal language. By using the body as an apparatus to decode, one can obtain information about the "lexicon of whiteness" through an individual's "[disposition] of Niceness versus Kindness" and "microaggressions versus microaffirmations". After deciphering the coded language of Niceness, Whisper Care can then operate through whispers, an act "to speak softly... especially to avoid being overheard" [61], by sharing stories and information in private conversation, a documented strategy in whispering networks [30,33]. These initially shared "secret" cues and offerings can be generously shared as a form of protection or aggression towards someone, warranted or not. Understanding who and/or what is safe or unsafe requires deep listening and discernment. As critical scholars, the authors employed diverse approaches to decode and confront whiteness as a survival mechanism, complemented by the need to understand and decode the lexicon of whiteness.

### 4.2. To What Extent Can I Trust You?

After an initial understanding of who a person is, we asked, "To what extent can I trust you?" Here, we evaluated alignment with liberation principles, ensuring a shared commitment to dismantling oppressive structures. Simultaneously, we scrutinized interactions and decoded this information for the subtleties of Niceness versus Kindness, focusing on the invisible labor needed to decode it in daily institutional interactions and practices:

Ruby: During this phase, there is an embodiment, a gut that you have to continue to exercise as you're learning about the other person. As we learn more about their orientation, the lexicon, the words, the actions, we are decoding, using our bodies to basically understand if you are someone that we can work with, that I can be in a relationship that is ethical, respectful, meaningful to seed a different world.

As early-career scholars, we are located in a liminal space, where we occupy a position of power, but within the broader hierarchical system of academia, we are near the bottom without tenure and lack institutional protection. Additionally, we cannot escape interacting with institutional leaders and other stakeholders, so in our journeys, asking "To what extent can I trust you (the other person)?" is essential for our survival. This second question dives into the realm of varying degrees of trust, where decoding such information within the lexicon and embodiment of whiteness takes emotional and intellectual labor. Whisper Care, and more specifically, this question, can protect early-career tenure-track faculty from marginalized backgrounds from microaggressions, gaslighting, and exhaustion from navigating whiteness and the everyday institutional politics of Niceness in their institution. Identifying the extent to which an individual can trust another person is an essential ingredient, not only in providing a sense of safety and security, but also in moving forward in collaborations, movement building, and reimagining a more just academic alternate reality.

### 4.3. Can I Build with You?

After learning about a person and discerning the extent to which we could trust this individual, we shifted our last question to, "Can I build with you?" Rooted in reciprocity and interdependence, this phase aims to establish a system of care within educational spaces that focuses on building community to imagine a different academic world, which we have

done with one another. While the intellectual labor of researchers is often to deconstruct (e.g., theories, concepts, problems, etc.), this question offers a call to action for academics to *re*construct the processes and systems they critiqued and deconstructed in the first place. More importantly, "Can I build with you?" explicitly attends to exploring the possibilities of building, or *re*constructing, together an alternate reality in higher education institutions. Within this realm, microaffirmations are pivotal in creating a supportive community where individuals feel acknowledged and valued [62]. Our collective strategy, Kindness and Whisper Care, centers on critical care as embodied knowledge. Grounded in the wisdom that the body serves as a guide, we recognize the importance of diverse positionalities, each contributing unique perspectives on practices of care. Building together from our CAE, we created a counterspace to navigate hostile environments disguised as Niceness, which, in turn, provided us a space to authentically engage with our experiences and heal from past trauma experienced in academia. The overarching strategy is to cultivate communities and systems of care, sowing the seeds for a more inclusive and affirming academic world.

## 5. Discussion and Implications

Collaborative autoethnography created a space for us to collectively conceptualize, document, and articulate Whisper Care, where we wove in, between, and out of one another's thoughts. While our findings memorialize our journey in creating a counterspace [52], these findings also describe a resistance-based framework in navigating and confronting Niceness in academia. Our everyday acts of resistance are enacted through the politics of refusal—refusing the practices of the colonial and processes of the neoliberal university [49,63,64] through Whisper Care. We contribute to the literature with a specific call to move beyond a culture of Niceness [7] and towards re-envisioning an academic world oriented around Kindness. Our findings offer Whisper Care as a framework, operating as both a lens and tool that presents an opportunity to deconstruct current academic realities poisoned with inequities, while reconstructing an alternate academic world filled with possibilities, hope, and critical care. Using theory in the flesh as both theory and methodology, we make evident that bodies are material sites of knowledge that need to be treated as a legitimate well [42,43], overflowing with rich and raw information, that actively resist, bravely confront, and strategically navigate hostile campus climates coded in Niceness [5,24,44]. In the three questions we introduced, we not only describe the process in theorizing Whisper Care, but these questions legitimize our lived experience to share with others. In conceptualizing Whisper Care, we identify a mechanism of carework in the form of microaffirmations [62] such as microvalidations, microtransformations, and microprotections [65]. Whisper Care encompasses a space and strategies essential for establishing a shield against gaslighting and the exhaustion incurred while navigating whiteness and Niceness within academic spaces, particularly for early-career tenure-track faculty from marginalized backgrounds.

Our findings offer future directions for both research and practice. First, we encourage future researchers to conduct studies that explore how historically minoritized faculty navigate institutional Niceness using resistance-based frameworks that emphasize individual and collective agency and embodied knowledge. Our findings highlight how embodied subjectivities informed the ways three tenure-track faculty navigated the daily rhetoric and practices of Niceness through a counterspace rooted in carework. Future studies can expand and legitimize the language to articulate an orientation and philosophy rooted in Kindness to combat Niceness in academia. Secondly, we urgently call on institutional leaders to actively listen to the experiences of their tenure-track minoritized faculty. While our findings present the ways we created a counterspace to support our professional journeys to tenure and life as scholars, creating an equitable environment is the onus of the institution. Actively finding members of an academic homespace and building a counterspace together, to be explicit, is an additional form of labor necessary for minoritized faculty to both survive and thrive in academia. As such, institutional leaders should think about building infrastructures through the lens of intersectionality that systematically supports the diverse

and evolving needs of minoritized early-career faculty. Lastly, we encourage minoritized faculty to actively seek out and create communities both in and outside their institution, which can be beneficial for early-career scholars. Our findings reveal the life-giving energy that community and collective care offers for early-career faculty, and the possibilities of what this can look like in academia [66]. Ultimately, these findings ask us to confront the paradox of existing as minoritized faculty in academia: to exist, we must resist; to move forward, we must push back; to transform, we must not conform. We know this work would not be possible if it were not for the lineage of scholars who have provided us mental and physical models to refuse academic conventions. Our radical aspiration is to affirm others in similar positions to ours, and call on researchers and higher education leaders to answer the question, "What will it take to build with us and with one another?" in their future research and current practice.

**Author Contributions:** All authors contributed to the conceptualization, analysis, writing, reviewing, and editing of this paper. All authors have read and agreed to the published version of the manuscript.

**Funding:** This research received no external funding.

**Institutional Review Board Statement:** The study was conducted with the Declaration of Helsinki and approved by the Institutional Review Board of the University of Nevada, Reno (2143562-1, 2 January 2024).

**Informed Consent Statement:** Informed consent was obtained from all subjects involved in the study.

**Data Availability Statement:** The original contributions presented in the study are included in the article, further inquiries can be directed to the corresponding author.

**Conflicts of Interest:** The authors declare no conflicts of interest.

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
