# Peer review of "Confronting Mean Girls Niceness: Conceptualizing Whisper Care to Disrupt the Politics of Niceness in Academia"

_education, doi:10.3390/educsci14050473_

Round 1

Reviewer 1 Report

Comments and Suggestions for Authors

Dear authors, this paper is an important an much needed intervention! 

Here are some of my humble suggestions to help strengthen a few aspects of your excellent work: 

Colonial Implication of Contemporary Niceness

For a deeper exploration of the missionary work undertaken by Christian White women as another facet of the colonial condition of Niceness, please refer to "Tender Violence in US Schools: Benevolent Whiteness and the Dangers of Heroic White Womanhood" by Natalee KÄ“haulani Bauer.

2.2. Integration of Minoritized Faculty Experiences with Niceness

Consider establishing a more explicit connection between Minoritized Faculty Experiences and the overarching theme of niceness. These practices, often veiled as benevolent, encompass highly racist undertones in compelling invisible labor.

Line 144: Within the context of whisper networks at line 144, the manifestation of carework involves the practice of sharing stories among women (Babel, 2018) for sustenance and information exchange. It may be beneficial to provide a precise definition of carework here. Some scholars are steering towards conceptualizing care beyond 'kindness' and gendered notions, viewing it as a form of resistance and refusal. Embedding care in the politics of refusal and fugitivity among women opens avenues for radical empowerment in theory and praxis, altering the ethical implications significantly. The term "resistance-based frameworks," suggested at line 433, could be more robustly tied to the proposed ethics of care, differentiating it from White Woman sentimentality. For a more in-depth exploration of niceness, care, and refusal, refer to "Working with Theories of Refusal and Decolonization in Higher Education," edited by Petra Mikulan and Michalinos Zembylas.

Theory in the Flesh Section

The "Theory in the Flesh" section, particularly the Theory and Methodology subsection, requires further development. It lacks citations to key works on the flesh-as-method, notably Spillers' "Mama’s Baby, Papa’s Maybe." Spillers articulates that the socio-political order of the New World, with its human sequence written in blood, represents a scene of actual mutilation, dismemberment, and exile. Perhaps all that is needed is to create a 'conversation' with the Black and Native Feminist Studies.

I recommend exploring works by Denise Ferreira da Silva and Lethabo King, particularly "Some Black feminist notes on Native feminisms and the flesh" (2021).

Kindness and Historical Violence

Delve deeper into how 'kindness' is utilized and conceptualized in the paper. Historical instances where kindness has been used as a moral discourse to justify violence, especially against the Other (woman, slave, child, queer), should be explored, including instances invoking the Bible.

What is meant by “Affective methods”? Which affect theory is being employed?

3.3. Positionality Section

The "Positionality" section requires clarification. The connection between the author’s research interests and their positionality within various relations (race, gender, sexuality, ability, and transgenerational spatio-temporal ontologies) is somewhat obscure. While research interests are undoubtedly influenced by individual positionality, the concept of research interest preconditioning positionality may require additional explanation.

Line 291: Define the scope of "our shared humanity" at line 291. Specify who is included and excluded in the term "our," referencing the authors.

Provide examples of the “daily rhetoric and practices of Niceness” faced in academia for readers and institutional leaders, particularly those who may be White and find it challenging to recognize their complicity. Share these examples if it feels safe to do so.

It would be valuable to add a section on the ethical implications of your work for the future of academia as a colonial institution.

Comments on the Quality of English Language

Quality of English is good.

Reviewer 2 Report

Comments and Suggestions for Authors

This is a truly original, inspiring and excellent article. I would not change anything, and look forward to its published version, which I definitely will spread among colleagues and students.

Author Response

Dear Reviewer 2,

We thank you for your feedback, and more specifically, the encouraging words and lifting up this piece when it's published.

Sincerely,

Authors